# CropPlantHarvest: A 500 m annual dataset of crop planting and harvesting dates (2001-2024) of the U.S. Midwest

Yin Liu<sup>1</sup>, Chunyuan Diao<sup>1</sup>

<sup>1</sup>Department of Geography and Geographic Information Science, University of Illinois Urbana-Champaign, Urbana, IL 61801, USA

Correspondence to: Chunyuan Diao (chunyuan@illinois.edu)

**Abstract.** As key components of agricultural management, planting and harvesting schedules have strongly influenced crop production by defining the length of the crop growing season and shaping the environmental conditions crops experience. Accurate knowledge of these management data is crucial for enhancing crop yield estimates by capturing the timing of crop development relative to weather and soil conditions, assessing climate adaptation by tracking shifts in farming practices over time, and supporting agricultural carbon accounting. Yet, existing planting and harvesting date datasets are largely based on state-level statistics or rule-based calendars that overlook intra-regional variability and the influence of human decision-making. The absence of long-term, high-resolution planting and harvesting date information hinders our ability to reconstruct historical agricultural practices and assess their agronomic and environmental consequences. In this study, we introduce CropPlantHarvest, the first dataset of annual corn and soybean planting and harvesting dates across the U.S. Midwest at 500 m resolution from 2001 to 2024. Planting dates are estimated using CropSow, an integrative remotely sensed crop modeling system that aligns simulated crop growth trajectories with satellite observations to retrieve field-level planting dates. Harvesting dates are retrieved using the Normalized Harvest Phenology Index (NHPI), a novel index that integrates Normalized Difference Vegetation Index (NDVI) and near-infrared (NIR) reflectance to detect harvesting events by capturing the distinct spectral transition from senescent crops to exposed crop residues. Validation against USDA crop progress reports and field-level dataset demonstrates high accuracy of CropPlantHarvest, with a mean absolute error of approximately 5 days for both crop species. This large spatial and temporal dataset captures management-driven variability in crop season timing and duration, supporting improved modeling of crop yields, greenhouse gas emissions, and resource use. It could also serve as a benchmark for refining remote-sensing phenology products and evaluating the agro-environmental impacts of evolving crop management decisions. CropPlantHarvest is available at https://doi.org/10.5281/zenodo.16967482 (Liu and Diao, 2025).

**Keywords:** Agricultural management practices; Planting date; Harvesting date; Phenology; Remote Sensing

## 1 Introduction



Food security will be increasingly challenged in the coming years due to population growth, shrinking availability of arable land, shifting consumption patterns, and climate change (Beddington, 2010). By 2050, global food production will need to increase by 60% to feed an estimated 9.3 billion people. In light of these projected changes, adapting crop management practices is essential to improve production efficiency and meet future food demands (Challinor et al., 2014). Among all crop management practices, the timing of planting and harvesting are especially critical, as they define the start and end of the growing season and determine the environmental conditions crops experience during development (Baum et al., 2020; Liu et al., 2023, 2024). As human-determined decisions, planting and harvesting dates reflect adaptive responses to shifting climate conditions (Kusumastuti et al., 2016; Xu et al., 2019; Huang et al., 2019; Baum et al., 2020; Shew et al., 2020). Adjusting planting dates can reduce exposure to temperature extremes and mitigate yield losses, while harvesting at the optimal time preserves grain quality and minimizes losses from adverse weather, pests, and disease (Jain et al., 2016; Liu et al., 2023, 2024). Accurate information on planting and harvesting dates is essential for assessing climate adaptation. Such data also enhances crop yield estimation by aligning crop development stages with observed weather and soil conditions (Bolton and Friedl, 2013; Zhang and Diao, 2023; Guo et al., 2024; Pei et al., 2025), and they play a central role in agricultural carbon accounting by identifying the periods when key exchanges of carbon, water, and energy occur between the land surface and atmosphere (Delgrosso et al., 2005; Zhang et al., 2020). Reliable records of these management decisions are therefore indispensable for climate adaptation analysis, crop yield estimation, and greenhouse gas accounting, ultimately supporting more accurate modeling and sustainable agricultural management.

In response to this need, several global datasets (e.g., SAGE (Sacks et al., 2010), MIRCA2000 (Portmann et al., 2010), RiceAtlas (Laborte et al., 2017), and PSHW (Iizumi et al., 2019), GCPE (Mori et al., 2023)) have been generated to provide information on crop planting and harvesting. MIRCA2000 and RiceAtlas compile crop-specific planting and harvesting months at subnational scales by integrating regional databases, national reports, and expert knowledge. In contrast, SAGE, PSHW, and GCPE apply rule-based methods that relate long-term climate variables (e.g., temperature, precipitation, and growing degree days) to the likelihood of planting and/or harvesting events, producing global estimates at a 0.5° resolution. While these datasets have been widely used to support research and policy efforts in climate adaptation, crop modeling, and sustainable land use, the coarse spatial resolutions of these datasets limit their ability to capture fine-scale spatial heterogeneity in crop management practices. Furthermore, they provide rough crop calendars or climate-derived estimates rather than actual management dates, which limits their capacity to capture temporal variability in planting and harvesting activities across diverse farming systems (Liu et al., 2023). This underscores the need for long-term, high-resolution datasets on planting and harvesting dates to better represent spatiotemporal variability in agronomic practices and improve modeling accuracy.

To address the data gap, remote sensing has been increasingly employed to estimate planting and harvesting dates across large spatial and temporal scales by interpreting vegetation dynamics captured in Vegetation Index (VI) time series (Cao et al., 2024; Diao, 2020; Diao et al., 2021; Diao and Li, 2022; Gao et al., 2017; L. Liu et al., 2022; Moulin et al., 1997; Sacks and Kucharik, 2011; Schwartz et al., 2002; Shen et al., 2023; Wu et al., 2017; Zhang et al., 2003). Several phenology datasets have been produced following this remote sensing-based strategies, where phenometrics derived from VI time series, such as the start of season (SOS) and end of season (EOS), are applied to approximate planting and harvesting timing (Bolton et al., 2020; Luo et al., 2020; Nieto et al., 2021; Niu et al., 2022). Yet, these datasets face important accuracy limitations in capturing planting and harvesting dates, as the phenometrics extracted typically using the feature points in VI curves do not consistently align with the actual timing of these field operations. Planting and harvesting practices are human-managed decisions that are primarily driven by external factors such as weather conditions, soil trafficability, and machinery availability, which can shift their feature positions on the VI curve (Liu et al., 2023, 2024). Moreover, planting is particularly challenging to detect directly, as it occurs before vegetation emerges and thus falls within an off-season flat portion of the VI curve that provides little signal to indicate

the event. Harvesting presents similar challenges when crops remain in the field long after maturity, in which case the event appears on a VI plateau rather than at a distinct turning point, as over-mature crops exhibit VI values comparable to bare soil. These limitations reduce the reliability of existing remote sensing—based datasets for accurately characterizing planting and harvesting dates (Gao and Zhang, 2021).

In this study, we introduce CropPlantHarvest, the first crop management practices dataset (i.e., planting and harvesting timing) for corn and soybean across the U.S. Midwest, mapped annually at 500 m resolution for 2001–2024. Planting dates are derived using CropSow, an integrative remotely sensed crop modeling system that assimilates remotely sensed VI time series with crop growth trajectories simulated by the Agricultural Production Systems sIMulator (APSIM) to estimate field-level planting dates (Liu et al., 2023). Harvesting dates are estimated with the Normalized Harvest Phenology Index (NHPI), a novel index that integrates Normalized Difference Vegetation Index (NDVI) and near-infrared (NIR) reflectance to detect the distinct spectral transition from senescent crops to exposed crop residues that marks harvesting events (Liu et al., 2024). Validation against field-level dataset and United States Department of Agriculture (USDA) crop progress reports demonstrates high accuracy of CropPlantHarvest, with a mean absolute error of approximately 5 days for both crop types. As one of the world's most productive and intensively managed agricultural regions, the U.S. Midwest plays a pivotal role in global food supply, contributing roughly one-third of global corn exports and over one-quarter of global soybean exports (Kucharik and Ramankutty, 2005). Its vast and heterogeneous landscapes, diverse management practices, and high sensitivity to seasonal timing make it a natural laboratory for studying dynamic changes of crop planting and harvesting decisions over space and time. Understanding these dynamics here is particularly valuable because management strategies that succeed in such a diverse, climate-sensitive system can offer transferable insights for other agricultural regions worldwide. Moreover, the Midwest's long history of consistent monitoring through yield statistics, crop progress reports, and environmental records provides a unique opportunity to link detailed planting and harvesting timings with production trends and climate adaptation strategies. Previous research at aggregated scales has shown that corn yield gains over the past half-century have been partially driven by earlier planting and longer growing seasons (Butler et al., 2018). By exploring these trends at fine spatial resolution, our 24-year dataset could enable more precise analyses of localized management adaptations, supporting improved crop models, yield estimates/forecasts, and agro-environmental assessments.

#### 95 2 Material and method

## 2.1 Material

100

To support the generation and validation of the planting and harvesting date products in U.S. Midwest, we utilize a comprehensive suite of datasets, including satellite imagery, crop type maps, meteorological data, soil property data, and ground reference observations. The U.S. Midwest states include Illinois (IL), Indiana (IN), Iowa (IA), Kansas (KS), Michigan (MI), Minnesota (MN), Missouri (MO), Nebraska (NE), North Dakota (ND), Ohio (OH), South Dakota (SD), and Wisconsin (WI) (Fig. 1). Satellite imagery provides the time series of spectral reflectance needed to detect planting and harvesting events. Crop type maps identify the locations of corn and soybean fields. Meteorological and soil datasets supply environmental inputs for the crop growth model used in planting date estimation. Ground reference observations are used both to calibrate the crop growth model used in planting date estimation and to evaluate the accuracy of the final planting and harvesting date estimates.

### 105 2.1.1 Satellite data

We use the Moderate Resolution Imaging Spectroradiometer (MODIS) MCD43A4 Version 6 product, a daily nadir Bidirectional Reflectance Distribution Function (BRDF)-adjusted reflectance dataset with a 500-meter spatial resolution, to detect corn and soybean planting and harvesting dates of U.S. Midwest from 2001-2024. From this dataset, we derive two key time series: the NDVI to characterize crop growth dynamics, and the Normalized Harvest Phenology Index (NHPI) to capture harvesting progression. The MODIS product offers an effective balance of spatial resolution, temporal frequency, and historical coverage, making it well-suited for long-term monitoring of planting and harvesting events. Its daily acquisition enables the

detection of subtle reflectance changes during both early and late crop development phases, which are critical periods for identifying planting and harvesting dates.

# 2.1.2 Crop type data

We use the USDA Cropland Data Layer (CDL), an annually updated crop-specific land cover product at 30 m spatial resolution, to extract spectrally pure crop signals from pure MODIS pixels of our study region (Boryan et al., 2011). MODIS pixels are classified as "pure" when at least 90% of the 30 m sub-pixels within the 500 m MODIS footprint are assigned to the same crop type. This threshold minimizes spectral mixing from other land covers (e.g., different crops, fallow land, or non-agricultural areas), ensuring that the extracted reflectance time series represent the phenological characteristics of a single crop type with minimal contamination. The CDL provides high classification accuracy for both corn and soybean, with producer's and user's accuracies typically exceeding 90%. For years or regions without CDL coverage prior to 2008, we supplement with the Corn—Soy Data Layer (CSDL), which offers historical corn and soybean classifications from 2001 onward.

# 2.1.3 Meteorological data

To support the simulation of crop growth trajectories in the planting date estimation process, we incorporate daily meteorological variables from the Daymet dataset (Thornton et al., 2020). These variables include minimum temperature, maximum temperature, precipitation, and shortwave radiation, which are required inputs for the crop growth model. Daymet provides continuous, gridded records at 1 km spatial resolution across North America from 1980 to present. For 2024, when Daymet data is not yet available at the time of our analysis, we substitute equivalent records from the Parameter-elevation Regressions on Independent Slopes Model (PRISM) dataset, which offers high-quality, gridded climate data at 4 km spatial resolution for the contiguous United States. In addition to driving the crop model simulation, these meteorological variables are also used to examine the influence of weather conditions on temporal variability in crop planting and harvesting dates.

### 2.1.4 Soil data


In addition to meteorological variables, soil properties are incorporated to provide crop growth modeling inputs that control water and nutrient availability, root development, and other processes influencing crop development within the planting date estimation system. Soil variables, including hydraulic properties, pH, and organic matter content, are obtained from the Gridded Soil Survey Geographic (gSSURGO) database. gSSURGO is a 30 m resolution raster product derived from the vector-based SSURGO database, developed through extensive field sampling and laboratory analysis by the National Cooperative Soil Survey. These inputs are critical for accurately simulating crop development from planting through vegetative growth within the system.

## 140 2.1.5 Ground truth data

Two sources of ground truth datasets are used for model calibration and/or product validation. The first is the USDA Crop Progress Reports (CPRs), which provide weekly state-level cumulative percentages of crops reaching specific phenological stages (e.g., planting and harvesting). CPR data are used to calibrate the crop growth model in the planting date estimation system by aligning simulated planting date distributions with reported planting progress. CPRs also serve as a reference for evaluating the temporal consistency of planting and harvesting date estimates at the state level. The second source is Beck's dataset, which contains field-level planting and harvesting records provided by Beck's Hybrids and is publicly available at <a href="https://www.beckshybrids.com/">https://www.beckshybrids.com/</a>. To account for the resolution difference between our 500 m product and Beck's plot-level measurements, we select MODIS pixels with each falling entirely within the boundary of a crop field, filtering out those spanning multiple fields with PlanetScope imagery verified through visual interpretation. This yields a total of 187 field records from 2016 to 2024 (Fig. 1), including 151 corn records and 36 soybean records. These records are used to validate the accuracy of both planting and harvesting estimates at the field scale.

Figure 1. Spatial distribution of corn and soybean planting and harvesting date records from Beck's dataset (2016–2024) across 12 U.S. Midwestern states. Yellow points indicate corn fields, and green points indicate soybean fields.

# 2.2 Method





For each pure crop pixel, its MODIS NDVI time series serves as the foundation for estimating planting and harvesting dates in corn and soybean fields. Conventional remote sensing-based methods often fail to capture true planting and harvesting dates, as feature points on VI curves do not consistently correspond to the actual timing of these field operations. To overcome this limitation, we use CropSow for planting date estimation and the NHPI for harvesting date estimation. Specifically, the annual NDVI curve is divided at its maximum value into two segments. The pre-peak segment spans from the start of the year to the timing of maximum NDVI (typically June-July in the U.S. Midwest), which represents canopy development and supports planting date detection with the CropSow system. The post-peak segment extends from maximum NDVI through the end of the year, which represents canopy senescence and crop residue exposure and supports harvesting date detection with the NHPIbased method (Fig. 2). For planting date estimation, the CropSow system integrates a remote sensing-based phenological detection method with the APSIM crop growth model. The phenological detection method identifies Greenup, which indicates the onset of active crop growth near emergence, while APSIM simulates the duration from planting to Greenup by accounting for soil properties, weather conditions, and crop characteristics within the soil-crop-atmosphere continuum. The planting date is then determined by subtracting the simulated planting-Greenup duration from the satellite-observed Greenup date. For harvesting date estimation, the Normalized Harvest Phenology Index (NHPI), defined as the ratio of near-infrared (NIR) reflectance to NDVI, enhances spectral differences between senescent vegetation and exposed crop residues to signal harvest. A threshold-based approach is applied to the NHPI time series to detect the sharp transition that corresponds to harvesting date. By combining these two event-specific methods across all target pixels, we generate CropPlantHarvest, a consistent, largescale dataset of corn and soybean planting and harvesting dates for the U.S. Midwest. The product is evaluated at field and state levels and used to assess long-term spatiotemporal trends and meteorological drivers.

Figure 2. Workflow for estimating planting and harvesting dates from MODIS NDVI time series of pure crop pixels. The left branch shows the CropSow system for planting date estimation, while the right branch shows the NHPI-based method for harvesting date estimation.

## 2.2.1 Planting Date Estimation

With the pre-peak segment of the NDVI curve, the planting date is estimated using the CropSow system, which consists of three main components: time series pre-processing, phenological characterization, and estimation of the planting-to-Greenup








duration (Liu et al., 2023). The first pre-processing step involves outlier removal, gap interpolation, off-season peak removal, and curve fitting. Outlier removal is performed sequentially using a quality assurance (QA) filter, a spline filter, a median absolute deviation (MAD) filter, and a snow filter to eliminate implausible observations caused by snow cover, clouds, haze, or low illumination. The QA filter removes the observations flagged as poor quality in the satellite product, including those affected by cloud contamination and cloud shadows. The spline filter smooths the NDVI curve and removes observations with residuals exceeding the mean plus/minus three standard deviations (Migliavacca et al., 2011). The MAD filter targets the removal of sharp spikes by applying the median absolute deviation criterion (Papale et al., 2006). Finally, the snow filter excludes observations with a Normalized Difference Snow Index (NDSI) above -0.2, removing snow-contaminated data that may not be fully identified by the previous steps (Liu et al., 2024). The filtered observations are then gap-filled using linear interpolation. Beyond outliers, the NDVI time series may contain off-season growth cycles unrelated to the target crop, typically resulting from weeds, cover crops, or double cropping. Such cycles can distort the characterization of phenological metrics for the target crop. To address this, a seasonality filter is applied to remove the off-season crop growth cycle (Diao, 2020). This filter employs a smoothing spline algorithm that fits a piecewise polynomial curve to the data, preserving the main seasonal growth trajectory while suppressing short-term fluctuations. Growth cycles are then delineated by turning points (i.e., peaks and troughs) in the smoothed curve, and each cycle is evaluated to determine whether it represents the main crop growth cycle. The evaluation considers both the NDVI magnitude of each peak and its timing relative to crop calendars from CPRs, ensuring consistency with the expected growth period of the target crop. The peak NDVI of the target crop cycle is required to occur between three weeks before the start of the corn silking stage (or the soybean flowering stage) and the start of the maturity stage reported in CPRs. Any cycles with localized peaks outside this temporal window are classified as off-season noise, and the segments between their preceding and subsequent troughs are removed.

Following time series pre-processing, Beck's double logistic method is applied to fit the pre-processed pre-peak segment to ensure a consistent representation of crop phenological dynamics (Beck et al., 2006). Compared to other fitting methods such as asymmetric Gaussian, Savitzky–Golay, quadratic, and nonlinear spherical methods, the double logistic approach can better characterize vegetation with rapid phenological transitions and relatively short growing seasons without overestimating season length. The Beck's double logistic function (Eq. 1) includes six parameters:  $V_{base}$  (off-season NDVI),  $V_{max}$  (maximum NDVI),  $m_2$  and  $n_2$  (timing of inflection points for green-up and senescence, respectively), and  $m_1$  and  $n_1$  (rates of change at those inflection points). These parameters are estimated by minimizing the root mean square error (RMSE) between the fitted curve f(t) and the pre-processed NDVI time series. While only pre-peak observations are used for fitting at this stage, the full double logistic formulation is retained to preserve the functional continuity and physiological realism of the phenological profile. This approach primarily constrains parameters associated with the green-up phase (i.e.,  $V_{base}$ ,  $V_{max}$ ,  $m_1$  and  $m_2$ ), while senescence-related parameters (i.e.,  $n_1$  and  $n_2$ ) remain weakly constrained. The curvature-based phenometric extraction method is then applied to the fitted curve to characterize the Greenup phenometric, defined as the first local maximum in the curvature change rate (Beck et al., 2006).

$$f(t) = V_{base} + (V_{max} - V_{base}) * \left(\frac{1}{1 + e^{(-m_1 * (t - m_2))}} + \frac{1}{1 + e^{(-n_1 * (t - n_2))}} - 1\right)$$
 (1)

Lastly, the Agricultural Production Systems sIMulator (APSIM) is used to estimate the interval between planting and the remotely sensed Greenup phenometric, which informs planting date retrieval. APSIM simulates crop phenological development by accounting for plant dynamic interactions within the soil—crop—atmosphere continuum. Specifically, the crop undergoes germination and emergence after planting, eventually reaching the remotely sensed Greenup stage, which roughly corresponds to the V3 stage (three collared leaves) in corn or the V3 stage (third trifoliate) in soybean. In APSIM, germination is triggered by soil moisture conditions, while subsequent phenological development is driven by the accumulation of thermal time, expressed as daily degree days. The rate of thermal time accumulation is adjusted for water and nitrogen stresses







simulated by the SoilWater and SoilNitrogen modules, respectively. The total thermal time from emergence to remotely sensed Greenup, denoted  $tt\_emerg\_to\_Greenup$ , is assumed to be consistent across cultivars within each state and year. Given a specific  $tt\_emerg\_to\_Greenup$ , the planting date is inferred by iteratively adjusting the planting date until APSIM-estimated Greenup aligned with observed Greenup phenometric. The parameter  $tt\_emerg\_to\_Greenup$  is calibrated annually using state-level CPRs and 1,000 randomly sampled pure MODIS pixels, by minimizing the RMSE between APSIM-simulated and CPR-reported planting date distributions. All APSIM input variables (e.g., fertilizer and planting density) follow settings established in previous planting date estimation study (Liu et al., 2023). Once calibrated, APSIM is run for each target crop pixel with varying planting dates and localized soil and weather inputs. The planting date that yields a simulated Greenup consistent with the observed Greenup phenometric is assigned as the estimated planting date for that field.

### 2.2.2 Harvesting Date Estimation

With the post-peak segment of the NDVI curve, harvesting dates are estimated using the NHPI-based method, which comprises three components: time series pre-processing, harvest phenology index generation, and harvest signal extraction (Liu et al., 2024). Time series pre-processing involves outlier removal, gap interpolation, preparation of the near-infrared (NIR) time series, and determination of the harvesting window. Outlier removal follows the same procedure applied in the pre-peak preprocessing, while gaps are filled through linear interpolation. In parallel, the NIR time series is processed by removing low-quality observations flagged during outlier detection and filling the resulting gaps with linear interpolation, ensuring consistency with the NDVI time series. The harvesting window is defined as the period starting from the middle of senescence (MOS), which is the date when NDVI declines to 50% of its senescence-phase amplitude (the difference between peak and minimum NDVI), and extending two months thereafter. This window isolates harvest-related changes, minimizes noise from other phenological phases, and improves the accuracy and efficiency of following harvesting date detection.

The second component focuses on generating the Normalized Harvest Phenology Index (NHPI), which is designed for capturing harvesting event signals. With pre-processed NDVI and NIR time series, the Harvest Phenology Index (HPI) is first calculated as the ratio of NIR to NDVI (Eq. 2). The NHPI is then obtained by normalizing the HPI within the harvesting window to a 0–1 scale using its local minimum and maximum values (Eq. 3). This normalization can enhance comparability across fields with varying environmental conditions.

$$HPI(t) = \frac{NIR(t)}{NDVI(t)} \tag{2}$$

$$NHPI(t) = \frac{HPI(t) - HPI_{min}(t_{start}, t_{end})}{HPI_{max}(t_{start}, t_{end}) - HPI_{min}(t_{start}, t_{end})}$$
(3)

where  $HPI_{min}(t_{start}, t_{end})$  and  $HPI_{max}(t_{start}, t_{end})$  represent the minimum and maximum HPI values observed during the harvesting window [ $t_{start}, t_{end}$ ], respectively. Here,  $t_{start}$  is defined as MOS, and  $t_{end}$  is two months after MOS.

In the final component, a threshold-based method is applied to extract the harvest signal from the NHPI time series. The harvesting date is identified as the first date when the NHPI exceeds the threshold 0.6, which is calibrated using field-level ground truth data of our study region. By amplifying the spectral contrast between senescent vegetation and crop residues, the NHPI-based method effectively characterizes harvesting transitions and enables robust, spatially and temporally generalizable detection of harvesting dates without the need for further calibration.

### 2.2.3 CropPlantHarvest Dataset Validation

We evaluate the final product at both the field and state levels. At the field level, planting and harvesting dates are validated against ground-truth data from Beck's dataset, which covers the U.S. Midwest from 2016 to 2024. To account for the spatial resolution difference between our 500 m product and Beck's field-level records, we select MODIS pixels with each falling





entirely within the boundary of a crop field, filtering out those spanning multiple fields with PlanetScope imagery verified through visual interpretation. We quantify accuracy using Mean Absolute Error (MAE) and the coefficient of determination (R<sup>2</sup>) (Eqs. 4 and 5), based on comparisons between estimated and observed harvesting dates for corn and soybean.

$$MAE = \frac{1}{n} \sum_{i=1}^{n} |y_i - \widehat{y}_i| \tag{4}$$

$$R^{2} = 1 - \frac{\sum_{i=1}^{n} (y_{i} - \widehat{y_{i}})^{2}}{\sum_{i=1}^{n} (y_{i} - \widehat{y})^{2}}$$
 (5)

where  $y_i$  is the observed planting or harvesting date of the sample i,  $\bar{y}$  is the mean of all the observed planting or harvesting dates, and  $\hat{y}_i$  is the estimated planting or harvesting date of the sample i. n denotes the number of samples.  $\sum_{i=1}^{n} (y_i - \hat{y}_i)^2$  is the sum of squared errors, and  $\sum_{i=1}^{n} (y_i - \bar{y}_i)^2$  is total sum of squares.

At the state level, we assess the planting and harvesting date products separately by comparing their aggregated temporal distributions with state-level statistics from USDA CPRs. For both planting and harvesting, the estimated dates of all corn or soybean pixels within each state and year are aggregated to generate corresponding cumulative distributions, formatted to match the structure of CPR data. To evaluate consistency, we sample each cumulative curve at 5% intervals between the 20th and 80th percentiles and compute the MAE and R<sup>2</sup> to quantify agreement between the satellite-based estimates and CPR-reported values.

#### 275 2.2.4 Trend and Meteorological Driver Analysis

To investigate long-term patterns in human-determined planting and harvesting dates, we conduct two complementary analyses: (1) a spatiotemporal trend analysis and (2) a meteorological driver analysis. For the spatiotemporal trends, county-level mean planting and harvesting dates are derived annually from the CropPlantHarvest product. Spatial patterns are examined by relating county-level values to latitude of the corresponding county centroids, given that latitude is a key determinant of temperature, which in turn influences agricultural management decisions. Temporal trends are estimated by fitting ordinary least squares (OLS) regressions of planting or harvesting dates against year at the county level. Temporal trends in growing season length are further assessed by calculating the difference between harvesting and planting dates and estimating its change over time using the same OLS. Growing season length provides an integrative indicator of how crop management practices and climatic conditions jointly influence the duration of the cropping cycle. The resulting slopes, expressed in days per decade, are summarized at the state level to highlight regional patterns.

For the meteorological driver analysis, we evaluate the influence of climate conditions on spatiotemporal changes in planting and harvesting dates. County-level meteorological variables, including average of daily minimum temperature and maximum temperature, and total precipitation, are calculated from Daymet data for the planting window (i.e., March to May) and harvesting window (i.e., August to October). These variables are selected because temperature and precipitation are the primary climate factors regulating planting and harvesting decisions. Minimum and maximum temperatures influence soil warming and crop development rates, which determine suitable windows for planting and harvesting, while precipitation affects soil trafficability and field accessibility for machinery operations. We apply county- and year-fixed effects panel regressions to isolate the effects of these variables, with county-fixed effects controlling for time-invariant county characteristics and year-fixed effects accounting for growing conditions common to all counties in a given year. The panel regression model is specified as follows:

$$Y_{it} = \beta_0 T_{min,it} + \beta_1 T_{max,it} + \beta_2 P_{it} + \alpha_i + \beta_t + \varepsilon_{it}$$
(6)

where  $Y_{it}$  denotes the planting or harvesting date for county i in year t.  $\beta_0$ ,  $\beta_1$ , and  $\beta_2$  capture the effects of average daily minimum temperature ( $T_{min,it}$ ), maximum temperature ( $T_{max,it}$ ), and total precipitation ( $P_{it}$ ) during the planting or harvesting season for county i in year t, respectively.  $\alpha_i$  represents county-specific fixed effects.  $\beta_t$  denotes year-specific fixed effects.  $\varepsilon_{it}$  is the error term. We conduct separate regressions for corn and soybean, as well as for planting and harvesting timing, with variables expressed in both standardized and unstandardized forms to allow comparison of effect sizes.

## 3 Results



#### 3.1 Dataset Validation

#### 3.1.1 Field-level Validation

At the field level, CropPlantHarvest shows a strong agreement with reference planting and harvesting dates, demonstrating its accuracy in capturing the timing of crop management practices. For planting dates, the evaluation yields an R² of 0.46 and a MAE of 6.40 days across 151 corn fields, and an R² of 0.51 with a MAE of 6.31 days across 36 soybean fields (Fig. 3a). Most predictions fall within ±10 days of observed planting dates, indicating reliable detection of planting timing. Some of the observed variations may be attributable to spatial mismatches between field-level observations and the 500 m resolution of the MODIS input data. For harvesting dates, the results show greater alignment with reference data compared to planting dates, with an R² of 0.749 and a MAE of 6.43 days for corn (N = 151), and an R² of 0.517 with a MAE of 5.39 days for soybean (N = 36) (Fig. 3b). This is likely attributable to more distinct spectral transitions at harvesting compared to planting, as well as differences in the detection systems used for the two events.

Figure 3. Field-level validation of estimated planting dates (a) and harvesting dates (b) for corn and soybean in the U.S. Midwest from 2016 to 2024.



## 3.1.2 State-level Validation

Using 2024 as an example, the temporal cumulative distributions of the estimated planting and harvesting dates are compared with those derived from CPR data for corn and soybean across 12 states in the U.S. Midwest (Fig. 4). The comparison shows strong consistency, indicating that CropPlantHarvest effectively reproduces the temporal progression of planting and harvesting for both crops and across all states. The MAEs between the estimated and reported cumulative curves are consistently within one week. For corn, planting date MAEs are less than 4 days in 8 states, and harvesting date MAEs are less than 4 days in 10 states. For soybean, planting date MAEs are below 4 days in 9 states, and harvesting date MAEs are below 4 days in 10 states. Spatial variations in the cumulative distributions across states reflect heterogeneity in planting and harvesting schedules, which are shaped by differences in environmental and climatic conditions, as well as management practices. Additionally, the temporal gap between the planting and harvesting curves exhibits substantial differences across states, reflecting differences in cultivar selection (e.g., maturity length), shaped by local agronomic practices and climate.

Figure 4. Temporal cumulative distributions of estimated planting and harvesting dates compared with corresponding state-level CPR data for corn (a) and soybean (b) in 2024.


Across all states and years, the estimated planting dates show strong agreement with CPR statistics, with an R<sup>2</sup> of 0.83 and MAE of 4.30 days for corn, and an R<sup>2</sup> of 0.86 and MAE of 3.76 days for soybean (Fig. 5a). Estimated harvesting dates also align well with CPR data, achieving an R<sup>2</sup> of 0.90 and MAE of 4.50 days for corn, and an R<sup>2</sup> of 0.84 and MAE of 3.97 days for soybean (Fig. 5b). These results show improved accuracy over previous state-level planting and harvesting date estimates from remotely sensed phenology characterization studies using SOS and EOS phenometrics (Yang et al., 2020; Shen et al., 2022), which report MAEs exceeding one week. The improvement is likely due to the tailored design of the proposed systems to specifically capture human-managed planting and harvesting activities. For planting, the CropSow system detects the emergence of a visible signal in the VI curve and estimates the time lag between planting and that detected signal, overcoming the limitation that planting typically occurs during the flat portion of the VI curve. For harvesting, the NHPI-based system identifies spectral feature points that exhibit stronger and more consistent correspondence with actual harvesting dates than those obtained directly from VI time series, addressing the inconsistent relationships between VI-based metrics and harvesting activity. Overall, the field-level and state-level evaluations confirm the reliability of the CropPlantHarvest product in capturing planting and harvesting dynamics for both corn and soybean across spatial and temporal scales.

Figure 5. State-level validation of estimated planting dates (a) and harvesting dates (b) for corn and soybean in the U.S. Midwest from 2001 to 2024. For each state-year, validation points are sampled from the estimated cumulative distribution percentiles (20%–80%) at 5% intervals and compared with corresponding CPR data, yielding one point per sampled percentile for each state-year-crop combination.


## 3.2 Spatial-temporal Trends Analysis

The averaged planting and harvesting date maps from 2001 to 2024 reveal distinct spatial patterns for both corn and soybean fields across the U.S. Midwest (Fig. 6). Corn planting generally begins earlier in southern regions and progressively later at higher latitudes, reflecting temperature-related constraints that influence planting decisions. Soybean planting dates show greater uniformity, likely due to staggered planting relative to corn and weaker temperature dependence. Corn harvesting follows a similar north—south gradient, occurring earlier in southern areas and later in northern regions, largely shaped by its planting pattern. In contrast, soybean harvesting exhibits weaker latitudinal patterns. These high-resolution maps also capture substantial within-state variation not reflected in state-level CPRs. For instance, northern Illinois consistently shows later corn harvests than southern Illinois despite similar planting windows. Such fine-scale detail underscores the value of this CropPlantHarvest product for understanding both local and regional cropping dynamics.

Figure 6. Spatial patterns of annual averages of planting and harvesting dates during 2001–2024 for corn (a), and soybean (b).

To characterize spatial variability in corn and soybean planting and harvesting dates across the U.S. Midwest, we further analyze the relationship between planting or harvesting dates and geographic location. As shown in Fig. 7, corn planting dates exhibit a nonlinear increasing trend with latitude across the U.S. Midwest. This pattern is likely related to the influence of temperature on planting decisions. In lower-latitude areas (e.g., ~36–38°N), planting typically occurs between DOY 90 and

115, while in higher-latitude regions (above 42°N), planting shifts later, often occurring between DOY 120 and 140. Warmer spring conditions in the south accelerate soil warming and enable earlier attainment of thermal thresholds required for seed germination and crop establishment. These favorable conditions provide greater planting flexibility, allowing farmers to begin planting corn earlier in the season. In contrast, cooler spring temperatures at higher latitudes delay planting readiness, resulting in progressively later planting dates toward the north. Corn harvesting dates follow a segmented latitudinal pattern, characterized by a steeper increase in lower-latitude areas (south of 41.11°N) and a more gradual increase in higher-latitude regions. Harvesting dates range from approximately DOY 250 in southern counties to around DOY 310 in the north. This trend generally corresponds to the planting pattern, as earlier planting often leads to earlier harvesting. However, minor deviations between the two are likely driven by hybrid selection and management strategies. Variations in corn cultivars influence the timing of physiological maturity, which in turn affects harvesting dates across different latitudes.

Soybean planting dates display a more diverse latitudinal pattern compared to corn. In southern regions (below 40.34°N), soybean is often planted later than that in some northern areas, generally between DOY 120 and 160. This delay is likely due to double cropping systems involving winter wheat, where soybean planting is postponed until winter wheat harvest is completed. In contrast, in central and northern regions soybean planting dates remain relatively consistent across latitudes, typically within DOY 130–145, reflecting the limited adoption of double cropping and the flexibility of later planting relative to corn once low-temperature risks have passed. Soybean harvesting dates exhibit a relatively stable latitudinal pattern, with harvest typically occurring around DOY 290. This pattern may be attributed to cultivar selection adapted to local conditions, enabling crops to reach maturity at a similar time despite differences in planting dates. In addition, soybean development is further regulated by photoperiod besides temperature, which contributes to a more compact and uniform growth cycle across latitudes. These factors together likely contribute to the observed uniformity in soybean harvesting timing.

390

395

400

Figure. 7. Relationship between mean county-level planting or harvesting dates and latitude in the U.S. Midwest for corn (a) and soybean (b) from 2001 to 2024, with a fitted trend line illustrating the latitude-dependent variation in planting or harvesting timing.

Fig. 8 presents trends in planting dates, harvesting dates, and growing season lengths for corn and soybean across 12 U.S. Midwestern states from 2001 to 2024. Positive values indicate delays, while negative values represent earlier timing. In general, corn planting dates show delayed trends across all states, though the magnitude varies. These delays may be driven by increasingly wet spring conditions, greater weather variability, or more cautious planting decisions in response to early-season risks. In contrast, soybean planting dates exhibit more heterogeneous patterns. Delays in states such as Michigan and Missouri may be due to double cropping with winter wheat or shifts in local management practices, while earlier planting trends in states like Illinois and Nebraska may result from increased planting flexibility or a shift from strictly sequential planting (soybean after corn) to partially overlapping planting windows. Harvesting date trends are generally smaller in magnitude than planting trends. Corn harvesting dates have been delayed in several states, including Illinois, Kansas, Missouri, and Ohio, likely due to the adoption of longer-season hybrids. Other states show relatively stable harvesting timing. Soybean harvesting dates remain relatively stable across the region. As a result, the corn growing season appears shortened in most states, suggesting that delayed planting has been the dominant driver. In contrast, the soybean growing season remains relatively stable across most states, likely because synchronized shifts in planting and harvesting have offset one another. These results suggest that corn and soybean systems are both adapting dynamically to changing climatic and agronomic conditions. Additionally, these differences reflect site-specific management strategies shaped by human decisions, which are influenced by varying understandings of local climate, cropping systems, and operational constraints.

Fig. 8. The trends of planting dates, harvesting dates, and growing season lengths during 2001-2024 by crops and states.

# 3.3 Meteorological Drivers of Planting and Harvesting

Regarding planting timing, Table 1 summarizes the regression results from county- and year-fixed effects panel models relating spring (March–May) weather conditions to the timing of planting dates for corn and soybean. For corn, higher mean minimum temperatures during spring are significantly associated with earlier planting dates, with each 1 °C increase in minimum temperature advancing planting by about 0.85 days (p 

435

440

and soil conditions are sufficiently dry, so minimum temperature and total precipitation serve as the primary climatic constraints. The standardized coefficient for precipitation (3.35) is slightly greater in magnitude than that for minimum temperature (-2.84), suggesting that excess soil moisture may be a more limiting factor than temperature for timely corn planting. For soybean, higher mean minimum and maximum temperatures have comparable effects on planting dates. Each 1 °C increase in minimum temperature advances planting by about 0.38 days, while higher maximum temperatures are associated with changes of similar magnitude. Precipitation is significantly linked to delayed planting, with each additional millimeter in spring delaying planting by about 0.0318 days (p < 0.001). The standardized coefficient for precipitation (4.6) is substantially greater than those for temperature (around -1.3), indicating that excess soil moisture is a more important constraint than temperature for soybean planting. This pattern reflects the fact that soybean is typically planted after corn, when the frost-free period has already begun, so minimum temperature exerts less influence on planting decisions and soil moisture becomes the dominant factor. In general, soybean planting, similar to corn, tends to occur earlier in warmer conditions and later in wetter conditions.

Table 1. Summary of fitted coefficients for planting season weather variables in county fixed-effects panel regression models estimating county-level planting day for corn and soybean. The weather variables include mean minimum temperature ( $T_{min}$ ), mean maximum temperature ( $T_{max}$ ), and total precipitation (Precip) during March–May. Coefficient type indicates whether coefficients are based on non-standardized or standardized weather variables. Standard errors are shown in parentheses.

| Crop type                                                  | Corn                 |                      | Soybean             |                     |  |  |
|------------------------------------------------------------|----------------------|----------------------|---------------------|---------------------|--|--|
| Coefficient Type                                           | Non-standardized     | Standardized         | Standardized        | Non-standardized    |  |  |
| $T_{\min}$                                                 | -0.8515 (0.1482) *** | -2.8396 (0.4941) *** | -0.3834 (0.1583) *  | -1.3116 (0.5418) *  |  |  |
| $T_{max}$                                                  | 0.0540 (0.1127)      | 0.1940 (0.4050)      | -0.3483 (0.1303) ** | -1.2580 (0.4708) ** |  |  |
| Precip                                                     | 0.0232 (0.0009) ***  | 3.3492 (0.1264) ***  | 0.0318 (0.0009) *** | 4.6135 (0.1375) *** |  |  |
| Adjusted R <sup>2</sup>                                    | 0.721                | 0.721                | 0.546               | 0. 546              |  |  |
| * for $p < 0.05$ , ** for $p < 0.01$ , *** for $p < 0.001$ |                      |                      |                     |                     |  |  |

Regarding harvesting timing, Table 2 summarizes the regression results from county- and year-fixed effects panel models relating late-season (August-November) weather conditions to the timing of harvesting dates for corn and soybean. A 1 °C increase in minimum temperature is associated with a delay in harvesting of 2.14 days for corn (p < 0.001) and 2.01 days for soybean (p < 0.001). This is because, in the U.S. Midwest, higher minimum temperatures often indicate more humid conditions (Kunkel et al., 2013). Factors such as soil moisture, evapotranspiration from crops, and the insulating effect of humid air contribute to warmer nights, and this elevated humidity slows the dry-down process, ultimately delaying harvesting decisions. In contrast, a 1 °C increase in maximum temperature is associated with an advance in harvesting of 4.00 days for corn (p < 0.001) and 3.35 days for soybean (p < 0.001). This is likely due to faster crop dry-down under warmer daytime conditions. Each additional millimeter of precipitation during the harvest window is associated with a delay in harvesting of 0.01–0.02 days (p < 0.001) for both crops, as wet weather can impede field access and machine operation. Based on standardized coefficients, maximum temperature has a much stronger influence on harvest timing than minimum temperature or total precipitation. Overall, these results show the asymmetric influence of diurnal temperature patterns and seasonal rainfall on planting and harvesting timing, with distinct crop-specific responses across the two management phases.

Table 2. Summary of fitted coefficients for harvesting season weather variables in county- and year-fixed effects panel regression models estimating county-level harvesting day for corn and soybean. The weather variables include mean minimum temperature (T<sub>min</sub>), mean maximum temperature (T<sub>max</sub>), and total precipitation (Precip) during August–October. Coefficient type indicates whether coefficients are based on non-standardized or standardized weather variables. Standard errors are shown in parentheses.

| Crop type                                          | Corn                 |                       | Soybean              |                      |  |  |
|----------------------------------------------------|----------------------|-----------------------|----------------------|----------------------|--|--|
| Coefficient Type                                   | Non-standardized     | Standardized          | Non-standardized     | Standardized         |  |  |
| $T_{\min}$                                         | 2.1408 (0.1998) ***  | 5.0634 (0.4726) ***   | 2.0141 (0.1652) ***  | 4.7819 (0.3922) ***  |  |  |
| $T_{max}$                                          | -3.9971 (0.1803) *** | -11.2444 (0.5072) *** | -3.3498 (0.1717) *** | -9.4188 (0.4828) *** |  |  |
| Precip                                             | 0.0160 (0.0014) ***  | 1.4946 (0.1354) ***   | 0.0143 (0.0012) ***  | 1.3148 (0.1119) ***  |  |  |
| Adjusted R <sup>2</sup>                            | 0.711                | 0.711                 | 0.536                | 0.536                |  |  |
| * for p 








regions, and assess their impacts on yield to inform policy interventions aimed at enhancing food security and system resilience. In addition, as critical inputs for process-based and remote sensing-based models, high-resolution planting and harvesting dates enable more accurate crop yield estimation, carbon accounting, and monitoring of other management practices. For crop yield estimation, they improve calibration by aligning simulated phenological stages with observed management events, thereby reducing prediction uncertainty (Zhang and Diao, 2023; Luo et al., 2023; Pei et al., 2025). For carbon accounting, they define the active growing and off-season periods, enabling earth system models to more accurately estimate biomass accumulation and seasonal carbon exchange, thereby improving assessments of carbon dynamics in agricultural systems (Delgrosso et al., 2005; Zhang et al., 2020). For crop management monitoring, they constrain remote sensing analyses to specific time windows, thereby improving the accuracy of detecting practices such as irrigation scheduling (Xie and Lark, 2021), cover-crop timing (Zhou et al., 2022; Wang et al., 2023), and tillage detection (Y. Liu et al., 2022; Zhang et al., 2024). This temporal detail provides a robust basis for yield forecasting, carbon accounting, and management monitoring of the U.S. agriculture, and establishes a foundation for extending applications to other regions.

Despite its spatial and temporal specificity and high accuracy in capturing human-managed planting and harvesting dates, CropPlantHarvest has several limitations. In order to provide long-term coverage back to 2001, it relies on MODIS imagery at 500 m resolution, which limits its ability to capture fine-scale field variability in heterogeneous agricultural landscapes. Future work could leverage spatiotemporal image fusion (Yang et al., 2021; Lyu et al., 2025), integrating fine-spatial, coarse-temporal-resolution data (e.g., Landsat) with fine-temporal, coarse-spatial-resolution data (e.g., MODIS), to generate long-term satellite observations with both fine spatial detail and frequent temporal coverage, thereby enabling more precise estimation of planting and harvesting dates. This approach would preserve the historical record while adding detailed spatial information, thereby improving the resolution and reliability of detected cropland management practices. In addition, estimating planting dates with CropSow requires the calibration of the thermal time threshold in the crop growth model, which is used to simulate the duration between planting and the remotely sensed Greenup phenometric. The accuracy of this calibration can be influenced by the quality and representativeness of the calibration data. Biases may occur when regional calibration data (i.e., CPR) are derived from limited or spatially imbalanced samples, which may not adequately represent regional planting conditions. Future work that calibrates CropSow using denser, field-level planting observations would help reduce these biases and further improve estimation accuracy.

## 510 5 Conclusion

In this study, we present CropPlantHarvest, the first annual, 500 m resolution dataset of corn and soybean planting and harvesting dates across the U.S. Midwest from 2001 to 2024. CropPlantHarvest provides essential data for climate adaptation analysis, crop yield estimation, and greenhouse gas accounting, filling the long-standing gap in consistent, fine-resolution datasets for large-scale monitoring of crop management practices. It leverages an integrative remotely sensed crop modeling system (i.e., CropSow) to estimate planting dates and a novel spectral index (i.e., NHPI) to detect harvesting dates with MODIS imagery, enabling the characterization of fine-scale spatiotemporal dynamics of these management practices. Validation against Beck's field-level observations and CPR state-level statistics demonstrate high accuracy, with mean absolute errors generally around 5 days for both crop species. The high spatial and temporal resolution of CropPlantHarvest enables tracking of shifts in planting and harvesting strategies over two decades, offering new opportunities to assess the impacts of climate variability, management adaptation, and policy on agricultural systems. As a consistent, long-term, and large-scale dataset, CropPlantHarvest provides a valuable resource for applications ranging from crop yield modeling and agricultural carbon accounting to broader evaluations of food security and climate resilience.

## **Data Availability**






Our CropPlantHarvest dataset, which provides planting and harvesting dates for corn and soybean fields at 500 m spatial resolution across the U.S. Midwest from 2001 to 2024, can be accessed via Zenodo: https://doi.org/10.5281/zenodo.16967482 (Liu and Diao, 2025).

#### **Author Contributions**

YL and CD conceptualized the study and developed the methodology. YL conducted data curation and formal analysis, implemented the software, and generated the visualizations. CD secured funding, managed the project, provided supervision and resources, and contributed to writing – review and editing. YL prepared the original draft with contributions and revisions from all co-authors.

## **Competing Interests**

The authors declared that none of the authors has any competing interests.

### Acknowledgements

This study is supported partly by the National Science Foundation (2048068 and 2518299) and partly by the National Aeronautics and Space Administration (80NSSC21K0946). It is also supported by the Taylor Geospatial Institute GISCoR (Geospatial Institute Seed Grant Program to stimulate Collaborative Research) (113356).

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
