# Peer review of "CropPlantHarvest: A 500 m annual dataset of crop planting and harvesting dates (2001-2024) of the U.S. Midwest"

_Earth System Science Data, 2025_

## Author Comment (AC1)

**Reply to Reviewer #1' comments**

The authors proposed a 500m annual dataset of crop planting and harvesting dates from 2001 to 2024 of 12 U.S. Midwest states for corn and soybean, named CropPlantHarvest. Planting dates are derived using CropSow and harvesting dates are calculated with Normalized Harvest Phenology Index (NHPI) based on NDVI and NIR reflectance. Field- and state-level evaluation showed the performance of the CropPlantHarvest.

**Response**: We thank the reviewer for the clear summary of our work and for highlighting the strengths of the proposed CropPlantHarvest dataset. We appreciate these positive comments. Below, we respond to your detailed comments point by point (all the line numbers are based on the revised track-change version).

Strengths:

1. Planting and harvest dates are derived in the proposed CropPlantHarvest dataset for crop management.

2. Multiple evaluations and analyses are discussed to verify the performance of the proposed dataset.

3. NHPI is introduced to represent the harvest phenology.

Weaknesses:

1. CDL at 30m spatial resolution is utilized as crop type data, which is very different with the proposed product that has a resolution of 500m. How to ensure accuracy under significant resolution differences?

**Response**: Thank you for the question. To ensure robustness under the scale difference between the 30 m CDL and the 500 m product, we adopt a strict pixel-purity filtering strategy to identify MODIS footprints corresponding to target crop types. Specifically, a 500 m pixel is retained for planting and harvesting date generation only when ≥90% of its underlying 30 m CDL pixels agree on the same crop type. This conservative criterion minimizes mixed-pixel contamination and ensures that the reference label represents a spatially homogeneous agricultural signal at the MODIS scale. The description of this procedure has been added in Section 2.1.2 (Lines 115–127).

2. In "2.2 Method" section, how to define a "pure crop pixel"?

**Response**: Thank you for the question. In this study, a "pure crop pixel" is defined at the 500 m MODIS scale based on fractional compositions of crop types of underlying 30 m Cropland Data Layer (CDL) pixels. Specifically, a 500 m pixel is considered a pure crop pixel if at least 90% of the 30 m CDL pixels within the MODIS footprint are classified as the same target crop type (e.g., corn or soybean). This threshold is applied to reduce mixed-pixel effects and ensure that the extracted phenological signals predominantly represent a single crop type. The formal definition and implementation details have been clarified in Section 2.1.2 (Lines 115–127).

3.  For Figure 3, it is recommended to provide a description of the meaning of solid and dashed lines.

**Response**: Thank you for this suggestion. We have revised the caption of Figure 3 to clarify the meaning of the solid and dashed lines. In the updated figure, the solid line represents the 1:1 line, indicating perfect agreement between the estimated and actual day-of-year (DOY) values. The dashed lines denote ±10 days from the 1:1 line, namely 10 days difference between the estimated and actual DOY. This clarification has been added to the Figure 3 caption to improve interpretability.

[Figure]

Figure 3. Field-level validation of estimated planting dates (a) and harvesting dates (b) for corn and soybean in the U.S. Midwest from 2016 to 2024. The solid line represents the 1:1 relationship between estimated and observed dates, and the dashed lines indicate ±10 days difference between estimated and observed ones.

4.  Still in Figure 3, why is corn's R2 much lower for harvesting date than planting data? Is the model robust enough for generalization?

**Response**: Thank you for this comment. The planting- and harvesting-date estimates are derived from two method components that are designed to capture different crop management signals, and as a result, they exhibit different performance characteristics. Planting is typically associated with relatively subtle spectral changes, whereas harvesting corresponds to more abrupt phenological transitions related to senescence and residue exposure. These inherent differences influence how

reliably each event can be detected from MODIS time series. To evaluate robustness and generalizability, both planting- and harvesting-date methods are assessed through comprehensive validation at multiple spatial scales, including field-level comparisons and state-level evaluations across multiple years. Across these analyses, both methods demonstrate stable and reasonable performance, supporting their applicability beyond individual fields or years. This point is discussed in Section 3.1.

5. The full name of "DOY" is suggested to be add when it is mentioned for the first time in the article.

**Response**: Thank you for this suggestion. We have revised the manuscript to define "DOY" at its first occurrence. Specifically, the text in Line 365 has been updated from *"planting typically occurs between DOY 90 and 115"* to *"planting typically occurs between day of year (DOY) 90 and 115."*

6. Does the location information contribute to the estimation of planting and harvesting dates?

**Response**: Thank you for the question. Location information contributes differently to the estimation of planting and harvesting dates due to the distinct designs of the two methods. For planting date estimation, location information is required indirectly through the APSIM crop growth model, which uses spatially explicit inputs such as soil properties and weather conditions to simulate the duration between planting and satellite-observed Greenup. In this case, geographic location provides the necessary context for linking remote sensing–derived phenological signals with process-based crop growth simulation. In contrast, harvesting date estimation relies solely on the Normalized Harvest Phenology Index (NHPI) derived from MODIS spectral time series. This approach detects harvesting events based on abrupt spectral transitions associated with senescence and residue exposure and does not require explicit location-based inputs or ancillary environmental data. As a result, harvesting date estimation is driven primarily by temporal spectral dynamics rather than geographic information. We have clarified this point in the Section 2.2 (Lines 161–182) to improve transparency regarding the role of location information in the two components.

7. For Figures 6-7, there are not (a) and (b) labels.

**Response**: Thank you for pointing this out. We have revised the captions of Figures 6 and 7 to align with the figure panels.

8. For Figure 7, there is a "+-" in the first subplot.

**Response**: Thank you for pointing this out. We have corrected the "+-" symbol in the first subplot of Figure 7.

[Figure]

Figure 7. Relationship between mean county-level planting or harvesting dates and latitude in the U.S. Midwest for corn and soybean from 2001 to 2024, with a fitted trend line illustrating the latitude-dependent variation in planting or harvesting timing.

9. For Table 1, why are different p thresholds employed to calculate the fitted coefficients?

**Response**: Thank you for the question. The p-value thresholds (\* $p < 0.05$, \*\* $p < 0.01$, \*\*\* $p < 0.001$) are used to indicate different levels of statistical significance, following common practice in regression analysis. Reporting multiple thresholds allows us to distinguish different levels of statistical significance for the estimated relationships, while all coefficients are derived from the same regression models using a consistent estimation procedure. We have clarified this explanation in the caption of Table 1 and Table 2 to avoid confusion.

Table 1. Summary of fitted coefficients for early season weather variables in county fixed-effects panel regression models estimating county-level planting day for corn and soybean. The weather variables include mean minimum temperature ($T_{min}$), mean maximum temperature ($T_{max}$), and total precipitation (Precip) during January–May. Coefficient type indicates whether coefficients are based on non-standardized or standardized weather variables. Standard errors are shown in parentheses. Statistical significance is indicated using multiple p-value thresholds (\* $p < 0.05$, \*\* $p < 0.01$, \*\*\* $p < 0.001$) to distinguish different levels of statistical significance. Coefficients without symbols are not statistically significant.

| Crop type | Corn | | Soybean | |
|---|---|---|---|---|
| Coefficient Type | Non-standardized | Standardized | Non-standardized | Standardized |
| $T_{min}$ | -0.8515 (0.1482) \*\*\* | -2.8396 (0.4941) \*\*\* | -0.3834 (0.1583) \* | -1.3116 (0.5418) \* |
| $T_{max}$ | 0.0540 (0.1127) | 0.1940 (0.4050) | -0.3483 (0.1303) \*\* | -1.2580 (0.4708) \*\* |
| Precip | 0.0232 (0.0009) \*\*\* | 3.3492 (0.1264) \*\*\* | 0.0318 (0.0009) \*\*\* | 4.6135 (0.1375) \*\*\* |
| Adjusted $R^2$ | 0.721 | 0.721 | 0.546 | 0. 546 |
| \* for $p < 0.05$, \*\* for $p < 0.01$, \*\*\* for $p < 0.001$ | | | | |

Table 2. Summary of fitted coefficients for harvesting season weather variables in county- and year-fixed effects panel regression models estimating county-level harvesting day for corn and soybean. The weather variables include mean minimum temperature ($T_{min}$), mean maximum temperature ($T_{max}$), and total precipitation (Precip) during August–October. Coefficient type indicates whether coefficients are based on non-standardized or standardized weather variables. Standard errors are shown in parentheses. Statistical significance is indicated using multiple p-value thresholds (* $p < 0.05$, ** $p < 0.01$, *** $p < 0.001$) to distinguish different levels of statistical significance. Coefficients without symbols are not statistically significant.

| Crop type | Corn | | Soybean | |
|---|---|---|---|---|
| Coefficient Type | Non-standardized | Standardized | Non-standardized | Standardized |
| $T_{min}$ | 2.1408 (0.1998) *** | 5.0634 (0.4726) *** | 2.0141 (0.1652) *** | 4.7819 (0.3922) *** |
| $T_{max}$ | -3.9971 (0.1803) *** | -11.244 (0.5072) *** | -3.3498 (0.1717) *** | -9.4188 (0.4828) *** |
| Precip | 0.0160 (0.0014) *** | 1.4946 (0.1354) *** | 0.0143 (0.0012) *** | 1.3148 (0.1119) *** |
| Adjusted $R^2$ | 0.711 | 0.711 | 0.536 | 0.536 |
| * for $p < 0.05$, ** for $p < 0.01$, *** for $p < 0.001$ | | | | |

**Reply to Reviewer #2' comments**

The manuscript presents a valuable, novel, and well-executed dataset with clear societal and scientific impact. The methodology is sound, validation is robust, and the discussion contextualizes the work effectively. The manuscript requires major clarifications and improvements to enhance its clarity, accessibility, and technical rigor.

**Response**: We thank the reviewer for the positive assessment of our work and for recognizing the dataset's novelty, impact, and methodological contributions. In response to the comments regarding needed clarifications and improvements, we have revised the manuscript to enhance clarity, accessibility, and technical rigor, and address the comments point by point below (all the line numbers are based on the revised track-change version).

1. The development of a long-term, high-resolution (500m, annual) dataset for both planting and harvesting dates addresses a critical gap in agricultural remote sensing and management studies. The integration of the CropSow modeling system and the novel NHPI index is a significant methodological advance over traditional phenology-based approaches.

**Response**: We thank the reviewer for recognizing the significance of the dataset and the methodological contributions of CropSow modeling system and the NHPI index.

2. The use of "pure" MODIS pixels (≥90% CDL agreement) is a good practice to minimize mixed signals. However, the manuscript should more explicitly discuss the potential implications of this filtering. What percentage of the total agricultural area in the Midwest is excluded? Could this introduce a spatial bias towards larger, more homogeneous fields?

**Response**: Thank you for this valuable comment. To explicitly assess the implications of purity-based filtering and address potential spatial bias issue, we conducted a sensitivity analysis using CDL agreement thresholds ranging from 50% to 90% (Fig. S1). The analysis reveals a clear trade-off between estimation accuracy and spatial coverage: increasing the purity threshold leads to monotonic improvements in accuracy for both planting and harvesting date estimates, while progressively reducing the proportion of agricultural area retained due to the declining availability of homogeneous MODIS-scale pixels. At the ≥90% threshold adopted in this study, approximately 20% of the total agricultural area across the Midwest is retained. Although this filtering preferentially selects more homogeneous fields, these fields are widely distributed with sufficient spatial coverage remained across all major Midwest states to support robust calibration and downstream applications, including crop yield estimation, carbon accounting, and monitoring of other management practices. We therefore adopt the ≥90% threshold as a high-confidence subset that minimizes mixed-pixel effects while maintaining regional representativeness. We have added a detailed description of the threshold selection process in the Methods and Discussion sections (Lines 115–127, 529–544) and included Fig. S1 in the supplementary material.

[Figure]

Figure S1. Sensitivity of planting and harvesting date accuracy to the pure-pixel threshold for (a) corn and (b) soybean across the U.S. Midwest in 2024. Solid lines show mean absolute error (MAE, days) for planting and harvesting date estimates, while dashed lines indicate the corresponding spatial coverage of agricultural area (%).

3. The calibration of the tt_emerg_to_Greenup parameter using state-level CPRs and 1000 random pixels is appropriate. However, the process for ensuring this single, state-year-level parameter is representative of sub-state variability in soils, cultivars, and microclimate could be elaborated.

**Response:** Thank you for the question. The *tt_emerg_to_Greenup* parameter represents the intrinsic thermal time required for crops to progress from emergence to early vegetative development, corresponding approximately to the V3 stage (defined as Greenup). This thermal requirement is primarily determined by the number of collared leaves and the thermal time required for the development of an individual leaf (i.e., *phyllochron*). Given that the SOS metrics used in this study approach the corn V3 stage (three collared leaves), the thermal time requirement from emergence to the remote sensing–derived SOS can therefore be reasonably assumed to be governed by *phyllochron*. Experimental studies have demonstrated that crop *phyllochron* is broadly comparable across cultivars and is primarily regulated by environmental conditions, particularly temperature and irradiance (Birch et al., 1998; dos Santos et al., 2022). Accordingly, calibrating a state–year-level thermal time requirement from emergence to SOS allows the CropSow model to capture the dominant environmental influences on *phyllochron* in a parsimonious manner. Sub-state heterogeneity in soils and microclimate is explicitly represented within APSIM through spatially resolved soil and weather inputs, which govern soil water balance and nitrogen dynamics at the field scale. These local conditions modulate the realized rate of early

crop development, resulting in substantial within-state variability in the timing of Greenup across fields. Consequently, within each state–year, the calibrated *tt_emerg_to_Greenup* parameter serves as a shared baseline thermal requirement, while field-level soil properties and microclimatic conditions drive spatial variability in the realized Greenup timing across the landscape. We have expanded the explanation of this design choice in the revised manuscript (Lines 224–247).

Reference:
Birch, C.J., Vos, J., Kiniry, J., Bos, H.J., Elings, A., 1998. Phyllochron responds to acclimation to temperature and irradiance in maize. Field Crops Research 59, 187–200. https://doi.org/10.1016/S0378-4290(98)00120-8
dos Santos, C.L., Abendroth, L.J., Coulter, J.A., Nafziger, E.D., Suyker, A., Yu, J., Schnable, P.S., Archontoulis, S.V., 2022. Maize Leaf Appearance Rates: A Synthesis From the United States Corn Belt. Frontiers in Plant Science 13.

4. Defining the harvesting window from MOS (50% senescence) to MOS+60 days is logical. However, in regions or years with very late harvest or early frost, could this window be truncated? Is the t_end parameter ever adjusted based on ancillary data (e.g., first freeze date)?
**Response**: Thank you for your thoughtful comment. Yes, in regions where crop remains in the field for extended periods after physiological maturity, such as Brazil or parts of subtropical Asia, the default two-month harvesting window can be extended to three to five months to accommodate longer post-maturity field-drying periods. As for $t_{end}$, defined as the end of the harvesting window, it is empirically derived from Beck's field-level dataset across the U.S. Midwest, which shows that the interval between the middle of senescence (MOS) and observed harvesting dates consistently falls within a two-month range, reflecting regional harvesting practices. While $t_{end}$ is not dynamically adjusted using ancillary data (e.g., freeze dates) in the current implementation, the window length can be flexibly adjusted in regions with shorter or longer post-maturity field residence to accommodate different harvesting behaviors. We have clarified this point in the revised manuscript (Lines 258–262).

5. The threshold of 0.6 for NHPI is stated as being calibrated using field-level data. It would be helpful to see a brief sensitivity analysis (e.g., in supplement) showing how the MAE changes with thresholds around 0.6 (e.g., 0.55, 0.65). This demonstrates the stability of the chosen value.
**Response**: Thank you for the suggestion. We performed a threshold sensitivity analysis using Beck's dataset covering the U.S. Midwest (2016-2024) to evaluate how different NHPI thresholds affect the agreement between estimated and observed harvesting dates. As shown in Fig. S2, applying thresholds of 0.1 or 0.9 results in significantly poorer performance with higher MAE compared to the optimal threshold of 0.6. This is because harvesting events typically coincide with a rapid increase in NHPI, and thresholds at the extremes (0.1 or 0.9) tend to identify dates well before or after the actual harvest event. We have added the description of the threshold selection process in the Method 2.2.2 sections (Lines 271-276) and included Fig. S2 in the supplementary material.

[Figure]

Figure S2. Sensitivity analysis of the NHPI threshold for estimating field-level harvesting dates. Mean Absolute Error (MAE; y-axis) is evaluated across a range of NHPI thresholds (x-axis) for corn and soybean, based on Beck's dataset from U.S. Midwest (2016-2024). Results indicate that a threshold around 0.6 consistently achieves the lowest MAE, suggesting optimal agreement between estimated and observed harvesting dates.

6. The effort to select MODIS pixels entirely within a single Beck's field is excellent and mitigates scale mismatch. The resulting field-level MAEs of ~6 days are very good. However, the text should more directly acknowledge that some of the remaining error is inherently due to the 500m vs. field-scale discrepancy and possible geolocation errors.

**Response**: Thank you for highlighting this point. We agree that, despite our effort to minimize scale mismatch by selecting MODIS pixels entirely within individual Beck's fields, some residual error is unavoidable. We have revised the manuscript to more explicitly acknowledge that part of the remaining field-level error is inherently due to the discrepancy between the 500 m MODIS resolution and field-scale observations, as well as potential geolocation uncertainties. This clarification has been added to the Results section (Lines 331-333).

7. Figures 3 and 5 (scatter plots) are clear. Figure 6 effectively shows spatial patterns. However, Figure 6 could be more informative if it included a panel showing the standard deviation (interannual variability) of planting/harvesting dates alongside the mean.

**Response:** Thank you for the suggestion. Following your recommendation, we have added a new figure (Fig. S3) presenting the spatial patterns of interannual variability (standard deviation) in planting and harvesting dates for corn and soybean during 2001–2024. The figure shows that interannual variability differs between crops, with corn exhibiting greater variability in harvesting dates than in planting dates, while soybean shows more comparable variability between planting and harvesting. We have added a brief description of these results in Lines 376–378 and included Fig. S3 in the Supplementary Material.

[Figure]

Figure S3. Spatial patterns of interannual variability (standard deviation) in planting and harvesting dates, shown alongside their long-term means, for corn and soybean during 2001–2024.

8. The non-linear relationships with latitude are interesting. The discussion attributing soybean patterns to double-cropping in the south is plausible. Is there any data (e.g., from CDL on winter wheat prevalence) that could be cited to support this claim more strongly?

**Response:** Thank you for this suggestion. To better support the interpretation of the non-linear relationships with latitude for soybean, we added a reference documenting the prevalence of soybean–winter wheat double-cropping systems in southern regions, which provides further explanation for the observed soybean patterns. The reference has been added to the Discussion (Lines 401, 597–599).

Reference: Borchers, A., Truex-Powell, E., Wallander, S., Nickerson, C., 2014. Multi-cropping practices: Recent trends in double-cropping (Economic Information Bulletin No. EIB-125). U.S. Department of Agriculture, Economic Research Service. https://doi.org/10.22004/AG.ECON.262122

9. The observed trends (delayed corn planting, stable/shortened seasons) are intriguing and well-discussed. The analysis would be even more powerful if linked directly to the driver analysis. For example, could the panel regression model be run on detrended data to separate interannual weather effects from long-term technological/management trends?

**Response**: Thank you for the suggestion. To separate interannual weather effects from long-term technological and management trends, we conducted additional county fixed-effects panel regressions using detrended planting date and harvesting date series. The detrended results show comparable relationships between weather variables and planting timing/harvesting timing, confirming the robustness of the driver analysis. We have added the information in lines 451-452, 471-473 in the revised paper and table S1 and S2 in the Supplementary Material.

Table S1. Summary of fitted coefficients for early season weather variables in county- and year-fixed effects panel regression models estimating county-level planting day for corn and soybean. The models are applied to detrended planting-date series to isolate interannual weather effects from long-term technological and management trends. The weather variables include mean minimum temperature ($T_{min}$), mean maximum temperature ($T_{max}$), and total precipitation (Precip) during January–May. Coefficient type indicates whether coefficients are based on non-standardized or standardized weather variables. Standard errors are shown in parentheses. Statistical significance is indicated using multiple p-value thresholds (* $p < 0.05$, ** $p < 0.01$, *** $p < 0.001$) to distinguish different levels of statistical significance. Coefficients without symbols are not statistically significant.

| Crop type | Corn | | Soybean | |
|---|---|---|---|---|
| Coefficient Type | Non-standardized | Standardized | Non-standardized | Standardized |
| $T_{min}$ | -0.9641 (0.1350) *** | -3.2150 (0.4503) *** | -0.3680 (0.1486) * | -1.2590 (0.5085) * |
| $T_{max}$ | -0.0169 (0.1020) | -0.0608 (0.3668) | -0.5843 (0.1246) *** | -2.1104 (0.4500) *** |
| Precip | 0.0231 (0.0008) *** | 3.3338 (0.1219) *** | 0.0314 (0.0009) *** | 4.5661 (0.1356) *** |
| Adjusted $R^2$ | 0.357 | 0.357 | 0.306 | 0.306 |
| * for $p < 0.05$, ** for $p < 0.01$, *** for $p < 0.001$ | | | | |

Table S2. Summary of fitted coefficients for harvesting season weather variables in county- and year-fixed effects panel regression models estimating county-level harvesting day for corn and soybean. The models are applied to detrended harvesting-date series to isolate interannual weather effects from long-term technological and management trends. The weather variables include mean minimum temperature ($T_{min}$), mean maximum temperature ($T_{max}$), and total precipitation (Precip) during August–October. Coefficient type indicates whether coefficients are based on non-standardized or standardized weather variables. Standard errors are shown in parentheses. Statistical significance is indicated using multiple p-value thresholds (* $p < 0.05$, ** $p < 0.01$, *** $p < 0.001$) to distinguish different levels of statistical significance. Coefficients without symbols are not statistically significant.

| Crop type | Corn | | Soybean | |
|---|---|---|---|---|
| Coefficient Type | Non-standardized | Standardized | Non-standardized | Standardized |
| $T_{min}$ | 1.9442 (0.1197) *** | 4.6237 (0.2846) *** | 2.2683 (0.1525) *** | 5.3826 (0.3618) *** |
| $T_{max}$ | -2.9336 (0.0979) *** | -7.3856 (0.2464) *** | -3.6269 (0.1597) *** | -10.196 (0.4491) *** |
| Precip | 0.0197 (0.0018) *** | 1.1205 (0.1020) *** | 0.0133 (0.0011) *** | 1.2214 (0.1047) *** |
| Adjusted $R^2$ | 0.446 | 0.446 | 0.325 | 0.325 |
| * for $p < 0.05$, ** for $p < 0.01$, *** for $p < 0.001$ | | | | |

10. The fixed-effects panel regression is a suitable method. The interpretation of coefficients (e.g., minimum temperature delaying harvest due to humidity) is reasonable. To bolster causality, consider if leading/lagging variables were tested (e.g., does spring precipitation affect planting, or precipitation in the weeks immediately prior to the median planting date?). **Response**: Thank you for this helpful suggestion. To strengthen causal interpretation, we additionally tested panel regression models using weather predictors restricted to the pre-planting period (January–March and two months preceding the overall median planting date), thereby avoiding simultaneity with observed planting dates. As shown in Table S3 and S4, early-season precipitation is significantly associated with planting timing, with greater precipitation linked to later planting dates, consistent with agronomic expectations that wetter soils require additional drying time before field operations can begin. We have added this analysis to the Discussion (Lines 508–510) and included the corresponding Table S3 and S4 in the Supplementary Material.

Table S3. Summary of fitted coefficients for pre-planting weather variables in county- and year-fixed effects panel regression models estimating county-level planting day for corn and soybean. To strengthen causal interpretation, all weather predictors are restricted to the pre-planting period (January–March), ensuring temporal precedence relative to observed planting dates. The weather variables include mean minimum temperature ($T_{min}$), mean maximum temperature ($T_{max}$), and total precipitation (Precip) during January–March. Coefficient type indicates whether coefficients are based on non-standardized or standardized weather variables. Standard errors are shown in parentheses. Statistical significance is indicated using multiple p-value thresholds (* $p < 0.05$, ** $p < 0.01$, *** $p < 0.001$) to distinguish different levels of statistical significance. Coefficients without symbols are not statistically significant.

| Crop type | Corn | | Soybean | |
|---|---|---|---|---|
| Coefficient Type | Non-standardized | Standardized | Non-standardized | Standardized |
| $T_{min}$ | -0.5703 (0.1009) *** | -3.2150 (0.4503) *** | -0.3834 (0.1583) * | -1.3116 (0.5418) * |
| $T_{max}$ | 0.1910 (0.0794) * | 0.8457 (0.3517) * | -0.3483 (0.1303) ** | -1.2580 (0.4708) ** |
| Precip | 0.0155 (0.0012) *** | 1.4177 (0.1131) *** | 0.0318 (0.0009) *** | 4.6135 (0.1375) *** |
| Adjusted $R^2$ | 0.707 | 0.707 | 0.546 | 0.546 |
| * for $p < 0.05$, ** for $p < 0.01$, *** for $p < 0.001$ | | | | |

Table S4. Summary of fitted coefficients for pre-planting weather variables in county- and year-fixed effects panel regression models estimating county-level planting day for corn and soybean. The weather variables include mean minimum temperature ($T_{min}$), mean maximum temperature ($T_{max}$), and total precipitation (Precip) during the two months preceding the overall median planting date. Coefficient type indicates whether coefficients are based on non-standardized or standardized weather variables. Standard errors are shown in parentheses. Statistical significance is indicated using multiple p-value thresholds (* $p < 0.05$, ** $p < 0.01$, *** $p < 0.001$) to distinguish different levels of statistical significance. Coefficients without symbols are not statistically significant.

| Crop type | Corn | | Soybean | |
|---|---|---|---|---|
| Coefficient Type | Non-standardized | Standardized | Non-standardized | Standardized |
| $T_{min}$ | -0.2334 (0.1168) * | -0.7822 (0.3915) * | 0.0353 (0.1423) | 0.1035 (0.4164) |
| $T_{max}$ | -0.0499 (0.0813) | -0.1971 (0.3216) | -0.8081 (0.1089) ** | -2.5081 (0.3381) ** |
| Precip | 0.0425 (0.0013) *** | 3.1095 (0.0972) *** | 0.0418 (0.0014) *** | 3.2990 (0.1069) *** |
| Adjusted $R^2$ | 0.725 | 0. 725 | 0.548 | 0.548 |
| * for $p < 0.05$, ** for $p < 0.01$, *** for $p < 0.001$ | | | | |

11. The limitation regarding MODIS 500m resolution is appropriately noted, with a future outlook towards data fusion. This section could be slightly expanded to quantify the potential error. For example, in highly fragmented landscapes, what is the typical sub-pixel heterogeneity?

**Response**: Thank you for this insightful comment. To quantify the potential impact of MODIS 500 m resolution and associated sub-pixel heterogeneity, we conducted a sensitivity analysis using CDL agreement thresholds ranging from 50% to 90% (Fig. S1). This analysis provides an empirical measure of within-pixel land-cover homogeneity and therefore serves as a proxy for sub-pixel heterogeneity. The results show that as the purity threshold increases, accuracy improves consistently for both planting and harvesting date estimation, indicating that less sub-pixel heterogeneity improves phenological signal retrieval. At the 90% threshold adopted in this study, retained pixels represent approximately 20% of the total agricultural area across the Midwest, corresponding to MODIS pixels with high land-cover consistency and reduced sub-pixel mixing. At this threshold, a sufficient number of pixels remain across all major Midwest states to support robust calibration. Together, these results help quantify the magnitude of sub-pixel heterogeneity effects and further motivate the use of high-purity pixels as a conservative strategy, while data-fusion approaches remain a promising direction for extending spatial coverage in future work. We have added this discussion to the Discussion section (Lines 529–534) and included Fig. S1 in the supplementary material.

[Figure]

Figure S1. Sensitivity of planting and harvesting date accuracy to the pure-pixel threshold for (a) corn and (b) soybean across the U.S. Midwest in 2024. Solid lines show mean absolute error (MAE, days) for planting and harvesting date estimates, while dashed lines indicate the corresponding spatial coverage of agricultural area (%).

12. The Zenodo repository is provided. Excellent. To maximize utility, please ensure the data is provided in a widely accessible, cloud-optimized format (e.g., GeoTIFF or Zarr for each year/crop/variable) with clear, machine-readable metadata.

**Response**: Thank you for this suggestion. The CropPlantHarvest dataset is provided in GeoTIFF format on Zenodo (https://zenodo.org/records/16967482), which is a widely supported and cloud-friendly raster format. Each file is organized using a consistent naming convention that explicitly encodes the crop type, year, and variable, enabling straightforward machine-readable identification and batch processing.

13. The manuscript is generally well-written. There are a few minor grammatical hiccups and slightly long sentences, particularly in the Method section.

**Response**: Thank you for this comment. We carefully reviewed the manuscript to address minor grammatical issues and revised the Method section to improve clarity and readability by shortening long sentences and streamlining phrasing throughout.

14. The Abstract and Conclusion accurately summarize the work. The Conclusion could be slightly strengthened by reiterating the immediate applications enabled by this dataset, echoing the discussion.

**Response:** Thank you for this helpful suggestion. To strengthen the conclusion and more explicitly reiterate the immediate applications of the dataset, we added text emphasizing that CropPlantHarvest can serve as a benchmark for refining remote-sensing phenology products and evaluating the agro-environmental impacts of evolving crop management decisions. It also enables more accurate yield and carbon-cycle modeling and improved monitoring of management practices (e.g., tillage, cover cropping, and irrigation) through clear delineation of growing and off-season periods. This revision has been incorporated into the conclusion (Lines 555–559).